# Roles of microRNAs in Regulating Cancer Stemness in Head and Neck Cancers

**DOI:** 10.3390/cancers13071742

**Published:** 2021-04-06

**Authors:** Melysa Fitriana, Wei-Lun Hwang, Pak-Yue Chan, Tai-Yuan Hsueh, Tsai-Tsen Liao

**Affiliations:** 1Graduate Institute of Medical Sciences, College of Medicine, Taipei Medical University, Taipei 11031, Taiwan; melysa.fitriana@mail.ugm.ac.id; 2Otorhinolaryngology Head and Neck Surgery Department, Faculty of Medicine, Public Health and Nursing, Universitas Gadjah Mada, Yogyakarta 55281, Indonesia; 3Department of Biotechnology and Laboratory Science in Medicine, National Yang Ming Chiao Tung University, Taipei 11221, Taiwan; a85296658@ym.edu.tw; 4Department of Biotechnology and Laboratory Science in Medicine, National Yang-Ming University, Taipei 11221, Taiwan; 5Cancer Progression Center of Excellence, National Yang Ming Chiao Tung University, Taipei 11221, Taiwan; 6School of Medicine, Taipei Medical University, Taipei 11031, Taiwan; b101108138@tmu.edu.tw (P.-Y.C.); b101108149@tmu.edu.tw (T.-Y.H.); 7Cell Physiology and Molecular Image Research Center, Wan Fang Hospital, Taipei Medical University, Taipei 11696, Taiwan

**Keywords:** microRNA, cancer stem cell, stemness, head and neck squamous cell carcinoma

## Abstract

**Simple Summary:**

Head and neck squamous cell carcinomas (HNSCCs) are highly heterogeneous human malignancies associated with genetic and environmental factors. In HNSCCs, cancer stem cells (CSCs) provide the plasticity for cancer cell progression, metastasis, therapeutic resistance, and recurrence. During carcinogenesis, microRNAs (miRNAs) play important roles in regulating the maintenance and acquisition of cancer stem cell features. Therefore, in this review, we summarize the roles of miRNAs in regulating the cancer stemness of HNSCCs to provide potential therapeutic applications.

**Abstract:**

Head and neck squamous cell carcinomas (HNSCCs) are epithelial malignancies with 5-year overall survival rates of approximately 40–50%. Emerging evidence indicates that a small population of cells in HNSCC patients, named cancer stem cells (CSCs), play vital roles in the processes of tumor initiation, progression, metastasis, immune evasion, chemo-/radioresistance, and recurrence. The acquisition of stem-like properties of cancer cells further provides cellular plasticity for stress adaptation and contributes to therapeutic resistance, resulting in a worse clinical outcome. Thus, targeting cancer stemness is fundamental for cancer treatment. MicroRNAs (miRNAs) are known to regulate stem cell features in the development and tissue regeneration through a miRNA–target interactive network. In HNSCCs, miRNAs act as tumor suppressors and/or oncogenes to modulate cancer stemness and therapeutic efficacy by regulating the CSC-specific tumor microenvironment (TME) and signaling pathways, such as epithelial-to-mesenchymal transition (EMT), Wnt/β-catenin signaling, and epidermal growth factor receptor (EGFR) or insulin-like growth factor 1 receptor (IGF1R) signaling pathways. Owing to a deeper understanding of disease-relevant miRNAs and advances in in vivo delivery systems, the administration of miRNA-based therapeutics is feasible and safe in humans, with encouraging efficacy results in early-phase clinical trials. In this review, we summarize the present findings to better understand the mechanical actions of miRNAs in maintaining CSCs and acquiring the stem-like features of cancer cells during HNSCC pathogenesis.

## 1. Introduction

Cancer is responsible for about 30% of all premature deaths from non-communicable diseases (NCDs) in adults aged approximately 30–69 years [1]. In 2018, there were 18.1 million people who suffered from cancer worldwide, and 9.6 million of them died from cancer (around one in six deaths globally). In addition, 354,864 (2% of all cancer sites) of new cases were of the lip and oral cavity, accounting for 177,384 (1.9% of all cancer sites) deaths [1,2]. Head and neck squamous cell carcinomas (HNSCCs) are epithelial malignancies located in the oral cavity, nasal cavity, pharynx (nasopharynx, oropharynx, and hypopharynx), and larynx [3,4]. HNSCC subtypes include oral SCCs (OSCCs), laryngeal SCCs (LSCCs), nasopharyngeal carcinomas (NPCs), and oropharyngeal SCCs (OPSCCs) [5,6]. It should be noted that with tongue squamous cell carcinomas, an OSCC includes the anterior two-thirds of the tongue (anterior to the circumvallate papillae) and an OPSCC consists of the base (or posterior one-third) of the tongue [7]. The incidence of HNSCC continues to rise and is anticipated to increase by 30% with 1.08 million new cases annually by 2030 [2,8,9]. Risk factors for the malignant incidence of HNSCCs include tobacco use, alcohol abuse, and human papillomavirus (HPV) infection [3,6]. Signs and symptoms can manifest as a lesion in the nose, mouth, or throat; a lump or neck mass; and ear discomfort; and functional abnormalities such as difficulty swallowing and/or chewing are often found in the later stages of these diseases [3]. The tumor, node, and metastasis (TNM) staging system is used for clinical staging and as a basis for treatment choice [10]. Therapeutic approaches include surgery, radiotherapy, chemoradiotherapy, a combination of surgery with radiotherapy or chemotherapy, and a combination of surgery with adjuvant chemotherapy and radiotherapy. Chemoradiotherapy may be taken as adjuvant therapy in advanced stages [11]. Unfortunately, despite several treatment options being available, outcomes of HNSCC treatment remain poor, patients generally develop resistance, and, as a result, the five-year overall survival rates of HNSCC patients are approximately 40–50% [6,12]. Advanced approaches have been developed by applying immunotherapy or combined immunotherapy treatment to treat resistant and recurrent cases [13].

The major obstacle in cancer therapy is tumor heterogeneity. Cancer stem cells (CSCs) are small populations of cancer cells and are well-known for their association with cancer resistance, relapse, tumorigenesis, and poor clinical outcomes in HNSCCs, which has promoted the development of novel and effective therapeutic protocols for better clinical outcomes [5,14]. Therefore, targeting CSCs has become an attractive approach for potential strategies to treat HNSCCs [15,16]. The abnormal activation of signaling cassettes, genetic and epigenetic modification, and microRNA (miR or miRNA) regulation are central regulators of CSC malignancy [17,18,19]. miRNAs work as hub regulators to modulate cell functions by binding to multiple 3′-untranslated regions (3′-UTRs) of target messenger RNAs (mRNAs) and cause the translational inhibition and/or degradation of transcripts [19,20,21]. Therefore, in this review, we address the roles of miRNAs in regulating the cancer stemness of HNSCCs.

## 2. CSCs and Cancer Stemness

CSCs are a minor population of cells within tumor tissues with a tumor-initiating capacity [22] and stem-like features, including self-renewal [23,24] and asymmetric cell division (ACD) [25]. Under chemotherapy, the cycling rates of CSCs slow and they enter the G0 phase in order to survive, accounting for therapeutic heterogeneity [26,27,28]. Cancer patients with higher stem cell signatures present poorly differentiated histological properties and are associated with a worse clinical outcome [29]. CSCs are heterogeneous populations. In colorectal cancer (CRC) tissues, prominin-1 (CD133) is the first molecular marker used to isolate colorectal cancer stem cells (CRCSCs) [30,31]. The epithelial-specific antigen (ESA)(+)/CD44 molecule and Indian blood group (CD44)(+) CRCSCs are associated with tumor recurrence after chemotherapy [32]. Dipeptidyl peptidase 4 (CD26)(+) CRCSCs enriched from CD133(+)/CD44(+) cells drive tumor metastasis [33]. Leucine-rich repeat-containing G protein-coupled receptor 5 (Lgr5)(+) CRCSCs are considered to be responsible for liver metastasis [34]. CD24 molecule (CD24) and activated leukocyte cell adhesion molecule (CD166) surface antigens are often combined with CD44 or CD133 for the identification and separation of CRCSCs [35]. In HNSCC, CSCs are grouped in accordance with the expression of surface markers such as CD44+ and aldehyde dehydrogenase 1+ (ALDH1+). CD44 mediates the adhesion, migration, and metastasis of CSCs [36], while ALDH1 ameliorate oxidative stress under therapeutic regimens such as platinum, taxanes, and oxazaphosphorine [37,38].

Despite the fact that the origins of CSCs have been linked to genetic mutation, epigenetic alterations, and unrestrained signaling pathways for the normal stem cells and progenitor cells [39,40], CSC properties would be induced or maintained by inflammatory mediators. Inflammatory cytokines and chemokines secreted by CSCs, including interleukin (IL)-1, IL-4, IL-6, and IL-8, sustain CSC niches in an autocrine manner [41,42,43,44]. Besides, the expression of IL-8 promotes the migratory and tube-forming capacities of endothelial cells [44]. IL-6 is also involved in cancer metastasis [45]. IL-6 activates Janus kinase 1 (JAK1) and phosphorylates programmed death–ligand 1(PD-L1) and promotes PD-L1 protein stability [46]. CSCs also enhance PD-L1 expression to escape immune surveillance, thereby enriching the CSC subpopulation [47,48,49]. In addition to secretory proteins, CSCs create an immunosuppressive, pro-tumoral microenvironment by releasing CSC exosomes for cancer progression [50,51].

To target CSCs, researchers have focused on deciphering how cancer cells acquire stemness properties. The major mechanisms involve the expression of genes associated with early development and aberrant intracellular signaling activation. Activation of stemness regulators sustains the stemness properties of HNSCCs, including the MYC proto-oncogene, bHLH transcription factor (MYC), sex-determining region Y-box 2 (SOX2), Nanog homeobox (NANOG), Krüppel-like factor 4 (KLF4), octamer-binding transcription factor 4 (OCT4), high-mobility group AT-hook 2 (HMGA2), cytokines, and epithelial-to-mesenchymal transition (EMT) transcription factors (EMT-TFs) [52,53,54,55,56,57,58]. On the other hand, abnormal signaling activation in Notch, Wnt(wingless)/β-catenin, transforming growth factor-β (TGF-β), Janus-activated kinase/signal transducer and activator of transcription (JAK/STAT), nuclear factor-κB (NF-κB), and the sonic hedgehog (SHH) pathway maintains cancer stemness [59,60,61,62,63,64]. Therefore, the rationale for identifying combinatorial therapeutic strategies combating CSC is intriguing.

## 3. miRNAs

miRNAs are non-coding (nc) RNA components with approximately 21–23 nucleotides that bind to and repress complementary mRNA targets [21,65]. Previously, ncRNAs were only considered to be evolutionary junk, but emerging evidence has indicated that miRNAs have important cellular functional roles and act as post-translational regulators [21,65,66,67]. miRNAs control around 30% of human genes, and about half of those genes are tumor associated or situated in vulnerable loci [68,69,70]; other studies have even suggested that miRNAs can regulate the expression of more than 60% of human genes [71,72]. miRNA expression is modulated by several mechanisms, such as transcriptional control, epigenetic modulation, and post-transcriptional regulation [67,73,74]. On the other hand, the biogenesis of miRNAs can mainly be divided into six steps: (1) RNA polymerase II transcribes miRNA genes into primary (pri)-miRNAs in the nucleus [75], (2) intermediate precursor (pre)-miRNA is released by pri-miRNA after being processed by the Drosha/DiGeorge syndrome critical region 8 (DGCR8) complex [75,76], (3) pre-miRNA bonds to exportin-5 (Exp5)/ras-related nuclear protein (Ran)-guanosine 5′-triphosphate (GTP) complex and is transferred to the cytoplasm [77], (4) the Dicer/(HIV-1 transactivating response (TAR)) RNA-binding protein (TRBP)/PACT complex turns pre-miRNA into double-stranded (ds) RNA in the cytoplasm [78,79,80], (5) the miRNA duplex is released into single strands by helicase [81], and (6) the miRNA-induced silencing complex (RISC) is bound to the 3′-UTRs of target mRNAs via the seed region of miRNA and subsequently triggers inhibition of the translation or degradation of target mRNAs [82]. The seed region of an miRNA (also known as the seed sequence) is a short, conserved sequence at nucleotides 2–8 at the 5′ end of the miRNA [83,84,85]. Therefore, the miRNA target prediction tools rely on an algorithm with the thermodynamics-based modeling of RNA, i.e., RNA duplex interactions with comparative sequence analysis to evaluate the seed region matching to the mRNAs [86].

The squamous epithelium covering the oral mucosa and skin depends on epithelial stem cells for tissue renewal [87]. In the oral mucosa, the basal cell layer harbors the self-renewing stem cells and their immediate descendants, the transient amplifying progenitor cells, to produce expanded terminally differentiating cells [88]. The terminally differentiating cells then leave the basal layer and form the outer layers to maintain the oral mucosa integrity. Therefore, stem cells and the proper controls between the phase transition of stem cells and differentiating cells are critical to maintaining tissue homeostasis. Evidence has shown that miRNA expression patterns control the epithelium stem cells’ characteristics. For example, Peng et al. indicated that the *miR-103/107* family is highly expressed in the stem-cell-enriched limbal epithelium. The *miR-103/107* family regulates and integrates these stem cell characteristics, thereby sustaining tissue maintenance and regeneration [89]. Moreover, studies have also indicated that the epidermal-specific deletion of enzymes responsible for miRNA maturation, such as DICER, Drosha, and DGCR8, severely impairs the homeostasis and morphogenesis of the epidermis [90,91,92,93]. These results indicate that miRNA expression is critical for the proper development of the epidermis and oral mucosa. Moreover, emerging evidence also highlights the importance of miRNAs in regulating carcinogenesis and CSCs. Better characterizations of miRNAs in regulating the stemness features of CSCs will contribute to better cancer treatment strategies (Figure 1).

## 4. miRNAs in Regulating Cancer Stemness

CSCs maintain and acquire stemness features through complex mechanisms, including abnormal activation of oncogenes, cytokines, signaling pathways, and EMT-TFs, as mentioned in Section 2 above. Studies indicated that miRNAs that regulate cancer stemness mainly depend on post-translational regulation to modulate activation of those stemness-related factors. Several studies have proven that abnormal miRNA expressions can act as oncogenes, tumor suppressors, or dual-role regulators [94,95]. All of these data highlight the potential for targeting miRNAs to eradicate CSCs, and researchers are working on anti-miRNA drugs and are searching for diagnostic miRNAs [96,97,98]. miRNAs have been applied as biomarkers to determine cancer prognoses and diagnoses due to their stability [99,100,101]. Xia et al. indicated that various tumor mutational burden levels had different miRNA expression patterns in HNSCC patients [102], and correlations between miRNA prognostic values as applied to HNSCCs have generated significant interest among researchers [103,104,105].

### 4.1. miRNAs as Oncomirs

As oncomirs, miRNAs can act as oncogenic miRNAs that promote biological processes such as proliferation, migration, angiogenesis, invasion, EMT, and stemness [106,107,108,109,110,111]. Oncomirs regulate cancer stemness through targeting their downstream targets which results in activation of stemness-related factors and signaling pathways. Therefore, oncomirs were shown to enhance tumor initiation and progression by modifying CSC properties such as self-renewal, tumorigenesis, drug resistance, and signaling pathways in cancer [112,113,114,115].

Several mechanisms for the oncogenicity of HNSCCs can be affected by miRNA presence. For example, *miR-125a* enhances the proliferation, migration, invasion, and stemness maintenance in cancer cells via suppressing *p53* expression [116]. The overexpression of p53 makes cell viability significantly decrease and induces cell cycle arrest at the G0/G1 phase [116]. *miR-134* suppresses E-cadherin expression and promotes OSCC cell progression through targeting programmed cell death 7 (*PDCD7*) [117]. E-cadherin can suppress cancer stemness by regulating the expressions of pluripotent genes (*MYC*, *NESTIN*, *POU5F1*, and *SOX2*) via the activation of Wnt/β-catenin signaling [118]. On the other hand, by suppressing the expression of the WW domain-containing oxidoreductase (*WWOX*) gene, *miR-134* can trigger oncogenicity and metastasis in HNSCCs [119]. WWOX is a tumor stemness suppressor that reduces the self-renewal ability of CSCs, differentiation potential, in vivo tumorigenic capability, and multidrug resistance [120]. Consistently, the downregulation of WWOX was indicated to induce EMT, enhance stemness, and increase chemoresistance in breast cancer [121]. *miR-1246* confers tumorigenicity and affects cancer stemness in OSCC through suppressing cyclin-G2 (*CCNG2*) [122] CCNG2 has been shown to suppress EMT by disrupting Wnt/β-catenin signaling [123], which has been proven to be involved in the migration and invasion of OSCCs [124].

Protocadherins are cell–cell adhesion molecules. The loss of protocadherins may contribute to cancer development not only by altering cell–cell adhesion but also by enhancing proliferation and EMT via activating the Wnt signaling pathway [125,126]. With LSCC, Giefing et al. showed that protocadherin 17 (*PCDH-17*) acts as a tumor suppressor gene [127]. Inhibition of *miR-196b* can suppress cell proliferation, migration, and invasion abilities but promote apoptosis by targeting *PCDH-17* in LSCC cells [128]. Moreover, LSCC patients with low expression of *miR-196b* and high expression of *PCDH-17* were shown to have an increase in the 5-year survival rates [128]. *miR-19a* and *miR-424* inhibit the TGF-β type III receptor (*TGFBR3*), also known as β-glycan, which results in promoting the EMT of tongue squamous carcinoma cells [108]. Other studies have also indicated that *miR-19a* promotes migration and EMT in gastric cancer, CRC, and lung cancer [129,130,131].

Mitogen-activated protein kinase (MAPK) signaling cascades are critical signal pathways related to EMT, which promotes cancer cell progression and metastasis in CSCs [132]. *miR-106A-5p* facilitates a malignant phenotype by acting as an autophagic suppressor through targeting BTG anti-proliferation factor 3 (*BTG3*) and activates autophagy-regulating MAPK signaling in NPC [133]. MYC target 1 (*MYCT1*), a direct target gene of MYC, is a novel candidate tumor suppressor gene cloned from LSCC [134,135]. MYCT1 protein suppresses *miR-629-3p* expression by reducing specificity protein 1 (*SP1*) expression. SP1 is also a TF for *miR-629-3p*, and its suppression enhances the expression of *miR-629-3p*’s downstream target, epithelial splicing regulatory protein 2 (*ESRP2*). Taken together, MYCT1 protein suppresses the EMT of laryngeal cancer via the SP1/*miR-629-3p*/ESRP2 pathway [136] Previous studies have shown that oral CSCs switch from expressing the CD44-variant form (CD44v) to expressing the standard form (CD44s) during the acquisition of cisplatin resistance, which results in EMT induction [137] During the process, CD44s induces *miR-629-3p* expression, which inhibits apoptotic cell death under cisplatin treatment conditions and promotes cell migration in HNSCCs [138]. Therefore, *miR-629-3p* serves as a therapeutic target to reverse chemotherapy resistance. Altogether, the miRNAs as oncomirs that regulate the stemness process of HNSCC are summarized in Table 1.

### 4.2. miRNAs as Tumor Suppressors

In contrast, tumor suppressor miRNAs were found to suppress activation of stemness factors, thereby decreasing CSC populations and tumor progression. Studies indicated that expressions of tumor suppressor miRNAs were commonly reduced in tumor samples. Conversely, their corresponding oncogenic downstream targets were activated, thereby activating stemness factors and enhancing the ability of cancer cells to acquire stemness features.

#### 4.2.1. miRNAs in HNSCC

The miRNA *let-7* family controls normal cellular development and differentiation, and a reduction in *let-7* contributes to carcinogenesis via the upregulation of oncogenic downstream targets and stemness properties [99]. Therefore, members of the *let-7* family are considered to be tumor suppressors for various cancers [139]. Ten members of the human *let-7* family have been identified, i.e., *let-7a*, *let-7b*, *let-7c*, *let-7d*, *let-7e*, *let-7f*, *let-7g*, *let-7i*, *miR-98*, and *miR-202*, which share the same seed region sequence [140,141]. Expressions of *let-7* family members decrease in HNSCCs patients, and among them, *let-7i* has been shown to most significantly suppress the expression of the chromatin modifier AT-rich interacting domain 3B (*ARID3B*). By suppressing *let-7i* expression, cells enhance ARID3B expression and acquire stemness features by activating embryonic SC (ESC)-specific genes such as *POU5F1*, *NANOG*, and *SOX2* via histone modifications [142]. The study also indicated that the EMT factor twist family bHLH transcription factor 1 (Twist1) cooperates with B lymphoma Mo-MLV insertion region 1 (BMI1), suppresses *let-7i* expression, and contributes to stem-like properties, thus enabling mesenchymal movements [143]. In OSCC of the tongue, ALDH1+ cells with cancer stemness characteristics show decreased expression of *let-7a* and high expressions of *NANOG* and *POU5F1*. *let-7a* overexpression in ALDH1+ cells further inhibited tumor formation and metastasis in vivo, suggesting that the *let-7a* gene plays an important role in modulating tumorigenesis stemness of HNSCC cells [144].

Moreover, radioresistance poses a major challenge in HNSCC treatment, in which CSCs are relatively radioresistant owing to different intrinsic and extrinsic factors [145]. Evidence has indicated that miRNAs might regulate not only stemness properties but also radiotherapy response. For example, *let-7c* contributes to oral cancer stemness and radio/chemoresistance through suppressing *CXCL8* (IL-8) [146]. Similarly, *CXCL8* was identified as a direct target of *miR-203*, and the reduction in *miR-203* promoted radioresistance by activating IL-8/AKT serine/threonine kinase 1 (AKT) signaling in NPC cells [147]. The low expression of *miR-203* was also showed to enhance EMT and result in intrinsic radioresistance of HNSCC, which could enable identification and treatment modification of radioresistant tumors [148]. *miR-520b* attenuates cell mobility via EMT suppression and suppressed spheroid cell formation, as well as reduced expressions of multiple stemness regulators (Nestin, Twist1, NANOG, OCT4) through targeting suppression of *CD44* in HNSCC cells [149]. Moreover, *miR-520b* also sensitized cells to therapeutic drugs and irradiation through targeting *CD44* [149]. CD44 is an adhesion molecule expressed in CSCs, which interacts with a glutamate–cystine transporter and controls the intracellular level of reduced glutathione (GSH). Therefore, CSCs with high CD44 expression show an enhanced capacity for GSH synthesis, resulting in higher reactive oxygen species (ROS) defense and radiotherapy resistance [150,151]. Therefore, *miR-520b* suppresses *CD44* and not only inhibits cancer stemness and multiple malignant properties but also sensitizes cells to chemoradiotherapy [149].

*miR-101* acts as a potent tumor suppressor, and its downregulation is associated with oral carcinomas [152]. In HNSCCs, low expressions of *miR-101* upregulate the oncogene Zeste homolog 2 (*EZH2*), which subsequently downregulates another tumor suppressor gene *rap1GAP*, which promotes HNSCC progression. EZH2 is a histone methyltransferase that belongs to the polycomb repressive complex 2 (PRC2) family that facilitates the trimethylation of H3K27 on the *rap1GAP* promoter to suppress its activation [153,154]. EZH2 can regulate cancer stemness by mediating the NOTCH1 activator and signaling to promote the initiation and growth of SCs [155]. EZH2 was shown to promote cell migration, invasion, and metastasis, and EMT, thereby enhancing cellular plasticity for oral tongue squamous cell carcinomas [156,157]. The *miR-29* family is also significantly downregulated in HNSCC patients [158]. Moreover, *miR-29b* suppresses DNA methyltransferase 3 beta (*DNMT3B*), resulting in inhibited EMT and promoted invasiveness of HNSCC cell lines through restoring E-cadherin expression by the demethylation of the promoter region [159].

*miR-204-5p* is a tumor suppressor in HNSCCs, which inhibits tumor growth, metastasis, and stemness by suppressing the signal transducer and activator of transcription 3 (STAT3) signaling and EMT via targeting *SNAI2, SUZ12, HDAC1*, and *JAK2* [160]. *STAT3* is a critical regulator of CSCs because of its relationship with EMT as one of the major proposed mechanisms for generating CSCs. It also plays a critical role in the angiogenesis and regulation of the tumor microenvironment (TME), which provides signals for differentiation or proliferation, especially through its involvement in the inflammatory NF-κB pathway [161]. *miR-124* was observed to target *STAT3* to repress tumor growth and metastasis in NPCs [162]. *miR-365-3p* targets the ETS homologous factor (*EHF*), a keratin 16 (KRT16) transcription factor, thereby suppressing KRT16 expression. The decrease in KRT16 further enhances the lysosomal degradation of β5-integrin and c-Met, leading to inhibition of their downstream Src/STAT3 signaling. In OSCC cells, *miR-365-3p* decreases migration, invasion, metastasis, cancer stemness, and chemoresistance via inhibiting Src/STAT3 signaling [163].

#### 4.2.2. miRNAs in OSCC

In OSCC, *let-7d* was shown to function as a negative regulator of EMT and exhibited chemoresistant properties and silencing of enhanced mesenchymal, stem-like, and chemoresistant traits through suppressing *TWIST1* and Snail family transcriptional repressor 1 (*SNAI1*) expression [164]. *miR-98* acts as a tumor suppressor, which reduces tumor cell growth and metastasis through targeting the insulin-like growth factor 1 receptor (*IGF1R*) in OSCCs [165]. IGF1R is critical in the human embryonic niche for self-renewal and SC expansion and regulates SC maintenance in normal tissue processes [166,167]. Moreover, the IGF1R pathway is critical for EMT induction/maintenance and the expansion of cancer stem-like cells [167,168,169,170]. In HNSCCs, Leong et al. indicated increased epidermal growth factor (EGF) receptor (EGFR) and IGF1R expressions and phosphorylation, which increased the activation of downstream pathways in ALDH1+ cells compared to ALDH- cells. Importantly, treatment with EGFR and IGF1R inhibitors reduced SC fractions, implying that the IGF1R is critical for maintaining HNSCC CSCs [171].

*miR-139-5p* overexpression inhibits OSCC cell proliferation, in vitro mobility of OSCC, and the expression of WNT-responsive *MYC*, *CCND1*, and *BCL2* through suppressing CXC chemokine receptor 4 (*CXCR4*) [172]. MYC-related signaling regulates CSC chemotherapeutic resistance and CRC organoids [173]. In addition, *miR-139* triggers the apoptosis of an oral cancer cell line, Tca8113 cells, through the Akt signaling pathway [174]. Another study suggested that *miR-139-5p* suppresses the tumorigenesis process and OSCC cell mobility by targeting homeobox (HOX)-A9 (*HOXA9*) [175]. HOX genes can encode master regulatory TFs that regulate SCs during development in various cancers; *HOX4* and *HOXA9* were observed to upregulate expression of the SC marker ALDH1 and increase SC self-renewal [176]. Similarly, *miR-495* was observed to suppress EMT, proliferation, migration, and invasion and promote the apoptosis of CSCs by inhibiting the *HOXC6*-mediated TGF-β signaling pathway in OSCCs [177]. Other studies have also indicated that *miR-495* significantly inhibits cell proliferation, migration, invasion, and EMT through the *miR-495*/IGF1/AKT signaling axis or by targeting *NOTCH1* in OSCCs [178,179].

The *miR-34* family contains three members, *miR-34a*, *miR-34b*, and *miR-34c*, clustered on two different chromosomal loci on chromosomes 1p36.22 (*Mir34a*) and 11q23.1 (*Mir34b/c*) [180,181]. In OSCCs and OPSCCs, *miR-34a* is described as a regulator of SCs [182]. *miR-34a* was observed to be downregulated in HNSCC tumors and cell lines [183]. Sun et al. observed that CSC enrichment by a spheroid culture showed significant downregulation of *miR-34a* expression. Furthermore, the restoration of *miR-34a* significantly inhibited EMT formation of the CSC phenotype and functionally reduced clonogenic and invasive capacities in HNSCC cell lines [184]. During the EMT process, cancer cells acquire the ability for tumor metastasis, invasion, drug resistance, and recurrence, which are associated with CSC functions. Gregory et al. first indicated that *miR-205* and the *miR-200* family (*miR-200a*, *miR-200b*, *miR-200c*, *miR-141*, and *miR-429*) suppressed the EMT by targeting zinc finger E-box binding homeobox 1 (*ZEB1*) and Smad-interacting protein 1 (*SIP1*, also known as *ZEB2*) in breast cancer [185]. Similarly, the *miR-200* family was indicated to enhance EMT through a reciprocal feedback loop between the *miR-200* family and ZEB1 in HNSCCs [186,187]. Recent emerging evidence has indicated that the EMT process might not simply be divided into a dichotomous system but may actually be an EMT spectrum. The epithelial/mesenchymal (E/M) hybrid status provides plasticity for cells with mixed E and M characteristics [188]. Lu et al. devised a unique model of miRNA-based coupled chimeric modules to elucidate the core regulatory network that underlies the hybrid E/M status. In that model system, two double-negative feedback loops of *miR-34*/SNAI1 and the *miR-200*/ZEB mutually regulate the E and M phenotypes and the hybrid phenotype. *miR-200*/ZEB was indicated to act as the decision-making module for cancer cells to undergo partial or complete EMT [189,190].

*miR-22* inhibits phosphatidylinositol 3-kinase (PI3K)/Akt/NF-κB signaling via downregulating activators such as *S100A8*, platelet-derived growth factor (*PDGF*), and vascular endothelial growth factor (*VEGF*), which implies a tumor suppressor role of *miR-22* in tongue squamous cell carcinoma [191]. PI3K is well known as a regulator for stemness-related signaling, including RAS/mitogen-activated protein kinase (MAPK) [192,193], NF-κB [194,195], Wnt/β-catenin [196,197], and TGF-β [198,199,200,201,202]. The NF-κB pathway maintains stemness by regulating many tumor-promoting inflammation-related cytokines, like tumor necrosis factor (TNF)-α [203], IL-1 [204], IL-6 [205,206], monocyte chemoattractant protein 1 (MCP1) [207], cytochrome oxidase subunit 2 (COX2) [203], and inducible nitric oxide synthase (iNOS) [203,208]. Simultaneously, the NF-κB pathway downregulates the expression of matrix metalloproteinases (MMPs) to increase tumor cell invasion [209]. *miR-22* also targets the expression of node-like receptor (NLR) family pyrin domain-containing 3 (*NLRP3*) and suppresses OSCC cell growth, migration, and invasion [210]. The NLRP3 inflammasome was associated with the carcinogenesis and CSC self-renewal activation in HNSCC patients with upregulated expression of BMI1, ALDH1, and CD44 [211]. The overexpression of *miR-22* results in reduced cell viability and an increase in the OSCC cell apoptotic rate by targeting the Wnt/β-catenin signaling pathway [212]. Qiu et al. indicated that the downregulation of *miR-22* would result in the upregulation of CD147 in tongue squamous cell carcinomas [213]. CD147 is also known as an extracellular MMP inducer, which promotes tumor initiation and progression through NF-κB signaling and also mediates the TGF-β1-induced EMT in HNSCC cells [214,215]. Therefore, CD147 might be a potential prognostic and treatment biomarker for HNSCCs.

#### 4.2.3. miRNAs in LSCC

In LSCC, *miR-98* was shown significantly reduced in both clinical specimens and cell lines, and *miR-98* directly targeted HMGA2-POSTN signaling and then suppressed cell migration, metastasis, invasion, and EMT-TFs of SNAI1 and Twist1, as well as SC-like features [216]. Moreover, *miR-101* inhibited tumorigenesis progression by regulating the Wnt/β-catenin signaling pathway by directly targeting cyclin-dependent kinase 8 (*CDK8*) in LSCC [217]. *CDK8* plays an important role in regulating biological processes at the transcription level in the Wnt/β-catenin signaling pathway, and it is considered a CRC oncogene [218,219].

#### 4.2.4. miRNAs in NPC

*miR-139-5p* inhibits the proliferation, invasion, and migration of human NPC cells by modulating EMT [220]. EMT enhances cancer cell motility and dissemination, which led to the concept of migrating CSCs as the basis of metastasis [221]. Findings have demonstrated a direct molecular link between EMT and stemness, where EMT activators such as Twist1 can co-induce EMT and stemness properties, thereby linking the EMT and CSC concepts [188]. EMT plays an important role in tumor metastasis and recurrence, and thus it is closely related to CSC functions [222,223]. Moreover, *miR-139-5p* reduces cisplatin resistance in NPC cells [220].

*miR-488-3p* activates the p53 pathway through suppressing zinc finger and BTB domain-containing protein 2 (*ZBTB2*), a reader of unmethylated DNA that regulates embryonic stem cell differentiation, thereby inhibiting proliferation and inducing apoptosis in esophageal SCCs [224,225]. p53 is able to suppress *CD44*, which is a CSC marker and suppresses cellular plasticity [226]. In NPCs, *miR-372* promotes radiosensitivity by activating the p53 signaling pathway via the inhibition of PDZ-binding kinase (*PBK*) [227]. Moreover, p53 represses EMT by mediating *miR-200c* expression, which causes the inhibition the translation of *ZEB1* and *BMI1* [228]. By downregulating *ATF3* expression, *miRNA-488* suppresses cell invasion and EMT in tongue squamous cell carcinoma cells [229]. Taken together, miRNAs as tumor suppressors that regulate the stemness process are summarized in Table 2.

### 4.3. miRNAs as Pleiotropic Functions

Some miRNAs play dual roles in oncogenes and tumor suppression, depending on the specific cell/tissue context. This reflects the complexity of the miRNA–target regulatory network. For example, *miR-107* was observed to antagonize and degrade *let-7*. *miR-107* suppressed *let-7* expression and activated downstream oncoprotein expressions such as HMGA2 and RAS and enhanced the tumorigenic and metastatic potential of cancer cells [230,231]. In HNSCCs, a *miR-107* increment was found in patients with lymph node metastasis, suggesting an oncogenic role for *miR-107* [232]; however, *miR-107* was indicated to suppress the proliferation, invasion, and colony formation of cells in LSCCs via inhibiting the voltage-gated calcium channel subunit α2δ1 (α2δ1) (encoded by *CACNA2D1*) [233]. In non-small-cell lung cancer (NSCLC), α2δ1 also enhances radioresistance in cancer stem-like cells by enhancing the efficiency of DNA damage repair [234]. Those results indicate the pleiotropic functions of *miR107* in HNSCCs.

miRNAs also mediated the regulation of cytokines/chemokines and the TME that modulates the CSC signaling pathway and sustains the CSC niche for acquiring and maintaining CSC features. For example, downregulation of *miR-9*, *miR-542-3p*, and *miR-34a*, and significant upregulation of *miR-21* were shown in CD44-positive CSCs with increased IL-6 and IL-8 expressions via targeting of the CD44v6/NANOG/phosphatase and tensin homolog (PTEN) axis in oral cancer [235]. *miR-9* acts as a tumor suppressor microRNA in HNSCC, and its role seems to be mediated through CXCR4 suppression [236]. Studies have indicated that *miR-9* overexpression results in decreased cellular proliferation and inhibited colony formation in soft agars when targeting *CXCR4* in HNSCC cells [236]. Conversely, another study has also indicated that *miR-9* was expressed at high levels in patients with recurrent HNSCCs [237]. Similar results were also shown for breast cancer with *miR-9* acting as a tumor suppressor in breast cancer proliferation in the early stage of breast cancer, while, with a higher malignancy, *miR-9* plays an opposite role in the metastatic process [238]. Thus, *miR-9* was suggested to have dual roles in carcinogenesis. Taken together, miRNAs with pleiotropic functions in HNSCCs are summarized in Table 3.

## 5. Conclusions

miRNAs can function as cancer suppressors or oncogenes, or even exhibit dual roles during cancer development, depending on the different cancer types or tumorigenesis stage. miRNAs are critical to tumor initiation, progression, metastasis, EMT, and chemoresistance via regulating CSC functions. miRNAs regulate important EMT-TFs and signaling pathways and modulate the TME to sustain and enhance cancer stemness. Therefore, targeting CSCs through miRNA manipulation provides a therapeutic opportunity for managing metastatic diseases. Moreover, with an understanding of miRNAs during tumorigenesis, we can take advantage of miRNA stability and use it as a diagnostic marker for primary diagnoses and patient follow-ups. We can also monitor miRNA changes to predict therapeutic responses as a non-invasive detection method. Recent studies have indicated that exosomal miRNAs can be better sources of biomarkers due to their advantages in terms of their quantity, quality, and stability [239]. Ludwig et al. indicated that *miR-205-5p* was exclusively expressed in HPV(+) exosomes, whereas *miR-1972* was only detected in HPV(−) exosomes. These miRNAs emerge as potential discriminating HPV-associated biomarkers [240] Intriguingly, human papillomavirus 16 (HPV16) infection has been indicated to enhance CSC properties, including ALDH1 activity, migration/invasion, and CSC-related factor expression, and enhances tumor growth OSCC cells [241]. Whether tumor-derived exosomes (TEX)-miRNAs are also involved in regulating the recipient cell stemness is unclear. In contrast to the extensive studies for cellular miRNAs in regulating cancer stemness, TEX-miRNA knowledge is relatively limited.

Moreover, Huang et al. indicated that only 5.63% of miRNAs were detected in both cells and TEX, which implies that cells can selectively pack certain miRNAs into exosomes in OSCC cells [242]. Meanwhile, exosomes can be released by various cell types, such as cancer-associated fibroblasts (CAFs) [243], dendritic cells [244], B cells [245], T cells [246], and tumor cells [247]. For example, Li et al. indicated that *miR-34a-5p* was significantly reduced in CAF-derived exosomes in OSCC patients. CAF transfers *miR-34a-5p*-devoid exosomes to OSCC cells and results in promoting the proliferation and motility of OSCC cells by upregulating the downstream target AXL (encoding AXL receptor tyrosine kinase). Therefore, the *miR-34a-5p*/AXL axis promotes the proliferation, metastasis, and EMT of oral cancer cells through the AKT/glycogen synthase kinase (GSK)-3β (GSK-3β)/β-catenin/SNAI1 signaling cascade [248]. Consistently, the cellular *miR-34a* significantly inhibited EMT formation of the CSC phenotype in HNSCC cell lines [184]. Hence, the sources and biological functions of exosomal miRNAs warrant further research before using them for screening and surveillance.

Among the tumor suppressor microRNAs of HNSCCs, *miR-34* is the only one that has been used in a clinical trial applied to treat primary liver cancer, small-cell lung carcinomas, lymphomas, multiple myelomas, and renal cell carcinomas. In 2013, the first microRNA-associated therapeutic drug was tested in a clinical trial (NCT01829971), MRX34, a special amphoteric lipid nanoparticle filled with *miR-34* mimics. Although this phase I study provided a dose-dependent modulation of relevant target genes that provide a proof-of-concept for MRX34 application for cancer therapy, severe adverse events were reported in five patients in terms of experiencing serious immune responses [249,250]. Hence, leading up to the MRX34 phase 2 clinical trials, NCT02862145 for melanomas has been withdrawn [251]. Other clinical trials have mainly focused on observational studies to explore the prognostic value of miRNAs in HNSCCs, for example, the prognostic value of *miR-29b* in the tissue, blood, and saliva in oral squamous cell carcinomas (NCT02009852). The *miR-29* family has been used to investigate Twist1-mediated cancer metastasis in HNSCCs (NCT01927354) (Table 4). Further research is warranted to determine the molecular functions and mechanisms of cellular or exosomal miRNAs, as well as their potential as miRNA-based diagnostics and therapeutics for HNSCCs.

## Figures and Tables

**Figure 1 cancers-13-01742-f001:**
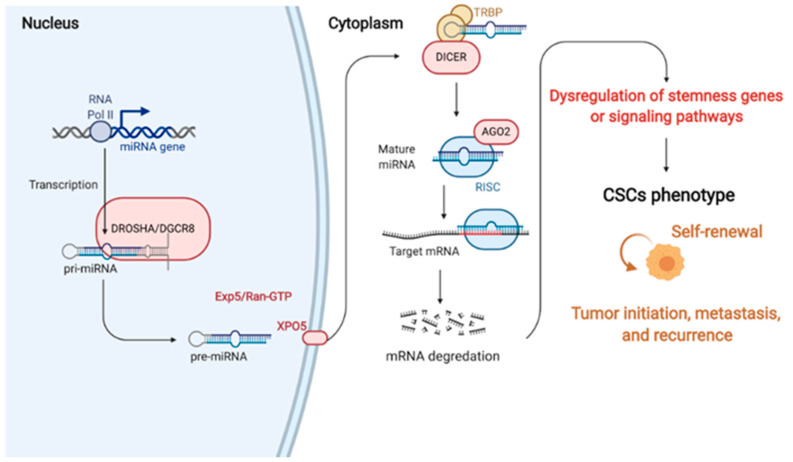
The generations and roles of miRNAs in cancer stem cell (CSC) regulation. In miRNA generation, miRNA genes are commonly transcribed by RNA Pol II in the nucleus. This transcription, which is cleaved by Drosha, produces a multiprotein complex with the DGCR8 protein. Cleavage induces pre-miRNA binding to the nuclear export factor EXO5 and then transported from the nucleus to the cytoplasm. In the cytoplasm, Dicer (another RNase III), which forms a complex with the double-stranded RNA-binding protein TRBP, cuts out the hairpin and produces an RNA duplex with approximately 22 nucleotides. The RNA duplex is dissociated to a single strand via AGO2 mediation, incorporated into the RISC, and binds to 3′UTR of the target mRNA to suppress the gene expression. Therefore, miRNAs can regulate cancer stemness properties and phenotypes by targeting the critical genes that control the activation of signaling pathways, transcriptional factors, and secreted factors. RNA Pol II, RNA polymerase II; pri-miRNAs, primary miRNAs; pre-miRNAs, precursor miRNAs; EXO5, exportin 5; AGO2, argonaute 2; RISC, miRNA-induced silencing complex; mRNA, messenger RNA; miRNA, microRNA (created with Biorender.com (accessed on 30 January 2021)).

**Table 1 cancers-13-01742-t001:** miRNAs as oncomirs that are involved in the stemness of head and neck squamous cell carcinomas (HNSCCs).

Oncomirs	Target(s)	Molecular Mechanism	Action Mode	Refs
*miR-125a*	*p53*	*miR-125a* enhances cell proliferation, migration, invasion, and stemness maintenance through suppression of *p53.*	Gene expression	[116]
*miR-134*	*PDCD7*	*miR-134* reduces E-cadherin expression by suppressing *PDCD7.* E-cadherin inhibition enhances expressions of pluripotent genes.	Gene expression	[117,118]
*miR-134*	*WWOX*	*miR-134* suppresses *WWOX*, a tumor stemness suppressor.	Suppressor inhibition	[119,120]
*miR-1246*	*CCNG2*	*miR-1246* promotes cancer stemness and tumorigenicity by suppressing *CCNG2.*	Suppressor inhibition	[122,123]
*miR-196b*	*PCDH-17*	*miR-196b* promotes cell proliferation, migration, and invasion abilities by inhibiting *PCDH-17.*	Suppressor inhibition	[127,128]
*miR-19a*, *miR-424*	*TGFBR3*	*miR-19a* and *miR-424* promote EMT by suppressing *TGFBR3.*	Signal transduction	[108]
*miR-106A-5p*	*BTG3*	*miR-106A-5p* inhibits autophagy and activates MAPK signaling by targeting *BTG3.*	Signal transduction	[133]
*miR-629-3p*	*ESRP2*	*miR-629-3p* enhances EMT via targeting *ESRP2.*	EMT process	[136]

EMT, epithelial-to-mesenchymal transition; MAPK, mitogen-activated protein kinase.

**Table 2 cancers-13-01742-t002:** miRNAs as tumor suppressors that are involved in the stemness of head and neck squamous cell carcinomas (HNSCCs).

Suppressors miRNAs	Target(s)	Molecular Mechanism	Action Mode	Refs
**HNSCC**				
*let-7i*	*ARID3B*	*let-7i* inhibition enhances *ARID3B* expression and activates the expression of *POU5F1*, *NANOG*, and *SOX2*.	Gene expression	[142]
*let-7a*	*NANOG, POU5F1*	Upregulation of *let-7a* in ALDH1^+^ cell suppresses tumor formation and metastasis.	Gene expression	[144]
*let-7c*	*CXCL8*	*let-7c* inhibition enhances stemness and radio-/chemoresistance by suppressing *CXCL8.*	Signal transduction	[146]
*miR-203*	*CXCL8*	*miR-203* reduction promotes EMT and activates IL-8/AKT signaling to trigger radioresistance.	EMT process/ Signal transduction	[147]
*miR-520b*	*CD44*	*miR-520b* inhibits EMT and the expression of stemness regulators and sensitizes cells to chemoradiotherapy through suppression of *CD44.*	EMT process/ Gene expression	[149]
*miR-101*	*EZH2*	*miR-101* inhibits *EZH2* and suppresses metastasis and EMT.	EMT process/ Signal transduction	[153,156,157]
*miR-101*	*CDK8*	*miR-101* inhibits *CDK8* expression and subsequently suppresses Wnt/β-catenin signaling and tumorigenesis.	Signal transduction	[217]
*miR-29b*	*DNMT3B*	*miR-29* suppresses *DNMT3B,* resulting in inhibition of EMT.	EMT process	[159]
*miR-204-5p*	*SNAI2*, *SUZ12*, *HDAC1*, and *JAK2*	*miR-204-5p* inhibits stemness by suppressing STAT3 signaling and EMT via targeting *SNAI2*, *SUZ12*, *HDAC1*, and *JAK2.*	EMT process/ Signal transduction	[160]
*miR-124*	*STAT3*	*miR-124* inhibits tumor growth and metastasis by suppressing *STAT3.*	Signal transduction	[162]
*miR-365-3p*	*EHF*	*miR-365-3p* decreases metastasis, stemness, and chemoresistance by suppressing EHF, which inhibits Src/STAT3 signaling.	Signal transduction	[163]
**OSCC**				
*let-7d*	*TWIST1, SNAI1*	*let-7d* suppresses EMT.	EMT process	[164]
*miR-98*	*IGF1R*	*miR-98* reduces self-renewal by suppressing *IGF1R*.	Signal transduction	[165]
*miR-139-5p*	*CXCR4*	*miR-139-5p* inhibits cell proliferation and the expression of WNT-responsive *MYC*, *CCND1*, and *BCL2* via inhibiting *CXCR4.*	Signal transduction	[172]
*miR-139-5p*	*HOXA9*	*miR-139-5p* inhibits *HOXA9*. *HOXA9* can increase stem cell self-renewal.	Gene expression/ Signal transduction	[175,176]
*miR-495*	HOXC6-mediated TGF-β signaling pathway	*miR-495* inhibits the HOXC6-mediated TGF-β signaling pathway and then suppresses EMT, proliferation, migration, and invasion.	Signal transduction	[177]
*miR-495*	IGF1/AKT signaling axis and *NOTCH1*	*miR-495* inhibits cell proliferation migration, invasion, and EMT through targeting the IGF1/AKT signaling axis or *NOTCH1*.	Signal transduction	[178,179]
*miR-34a*	EMT	*miR-34a* inhibits EMT formation.	EMT process	[184]
*miR-200* family	*ZEB1/2*	The *miR-200* family suppresses EMT by targeting *ZEB1/2.*	EMT process	[186,187]
*miR-22*	KAT6B	*miR-22* inhibits activators of PI3K/Akt/NF-κB signaling.	Signal transduction	[191]
*miR-22*	*NLRP3*	*miR-22* inhibits *NLRP3* which suppresses expressions of BMI1, ALDH1, and CD44.	Signal transduction	[210,211]
*miR-22*	*CD147*	*miR-22* inhibits *CD147*, which suppresses tumor initiation and progression through NF-κB signaling and mediates TGF-β1-induced EMT.	EMT process/ Signal transduction	[213,214,215]
**LSCC**				
*miR-98*	HMGA2-POSTN signaling	*miR-98* inhibits HMGA2-POSTN signaling, which suppresses metastasis and EMT-TFs.	EMT process/ Signal transduction	[216]
*miR-101*	*CDK8*	*miR-101* inhibits *CDK8* expression, which suppresses Wnt/β-catenin signaling and tumorigenesis.	Signal transduction	[217]
**NPC**				
*miR-139-5p*	EMT	*miR-139-5p* suppresses EMT and inhibits proliferation, invasion, migration, and cisplatin resistance.	EMT process	[220]
*miR-488-3p*	*ZBTB2*	*miR-488-3p* activates the p53 pathway by suppressing *ZBTB2* to inhibit proliferation and induce apoptosis.	Signal transduction	[224,225]
*miR-372*	*PBK*	*miR-372* activates the p53 signaling pathway via repressing *PBK* to promotes radiosensitivity.	Signal transduction	[227]

EMT, epithelial-to-mesenchymal transition; TFs, transcription factors; OSCC, oral squamous cell carcinoma; LSCC, laryngeal squamous cell carcinoma; NPC, nasopharyngeal carcinoma; IL-8, interleukin 8; EHF, ETS homologous factor; STAT3, signal transducer and activator of transcription 3; IGF1, insulin-like growth factor 1; AKT, AKT serine/threonine kinase 1; TGF-β, transforming growth factor-β; HMGA2, high-mobility group AT-hook 2; NF-κB, nuclear factor-κB.

**Table 3 cancers-13-01742-t003:** miRNAs with pleiotropic functions that are involved in the stemness of head and neck squamous cell carcinomas (HNSCCs).

Pleiotropic miRNAs	Target(s)	Molecular Mechanism	Action Mode	Refs
*miR-107*	*let-7*	*miR-107* suppresses *let-7* expression and activates downstream oncoprotein expressions for enhancing tumorigenic and metastasis.	Gene expression/ Signal transduction	[230,231]
*miR-107*	*CACNA2D1*	*miR-107* suppresses proliferation, invasion, and colony formation of LSCC cells via inhibiting *CACNA2D1*.	Signal transduction	[233]
*miR-9*	CD44v6/NANOG/PTEN axis	*miR-9* inhibits the CD44v6/NANOG/PTEN axis for suppressing IL-6 and IL-8 signaling.	Signal transduction	[235]
*miR-9*	*CXCR4*	*miR-9* decreases proliferation and colony formation by targeting *CXCR4*.	Signal transduction	[236]

LSCC, laryngeal squamous cell carcinoma; NANOG, Nanog homeobox; PTEN, phosphatase and tensin homolog.

**Table 4 cancers-13-01742-t004:** List of miRNA-related clinical trials in head and neck squamous cell carcinomas (HNSCCs).

miRNAs	Clinical Trials	Trial ID
*miR-29b*	Observational study to explore the prognostic value of *miR-29b* in tissue, blood, and saliva in OSCC	NCT02009852
*miR-29* family	Observational study to investigate the role of *miR-29* family in Twist1-mediated cancer metastasis in HNSCC	NCT01927354

OSCC, oral squamous cell carcinoma.

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
