# Peer review of "Roles of microRNAs in Regulating Cancer Stemness in Head and Neck Cancers"

_cancers, 2021, doi:10.3390/cancers13071742_

Round 1

Reviewer 1 Report

The authors summarized the latest finding regarding miRNAs that regulate HNSCC stemness in detail. They also introduced their targets and molecular mechanisms in cancer stem cell maintenance in HNSCC. The review will be very useful to the filed. If would be more helpful if the author can include any current clinical trials information about these miRNAs. This could be summarized in a table and included in the conclusion along with prespectives and more elaboration of what needs to be achived in the future. Besides, there were typos in lines 118 and 119. 

Reviewer 2 Report

This review article summarizes the functional impact of miRNAs in the regulation of cancer stemness in HNSCC. The article gives a good overview on numerous articles in that field.

However, there are some issues regarding the structure and content of the review: The review should be re-structured to improve readability. It might be beneficial to incorporate more subheadings or to re-structure according to HNCC subsides. From a clinical perspective:  The subdivision into "tongue SCC" doesn't make sense at all. The clinical and pathological TNM classification distinguishes SCCs from tongues into either oropharyngeal SCCs (base of tongue) or oral SCCs (rest of tongue). Line 60/61: This sentence is not correct as it is: Radiotherapy and chemotherapy alone are no therapeutic options for HNSCCs except for palliative treatment. Chemotherapy is usually combined with radiotherapy in an adjuvant setting. Table 1: miR-205-5p and miR-1972 are missing in the list and text. Lately, there has been evidence that these two could be exosomal markers for HPV+/- HNSCCs: miR-205-5p in HPV(+) and miR-1972 in HPV(-) exosomes (PMID: 33202950)

Formal issue: Genes should be italicized.

The authors should consider English editing services: There are some spelling issues (i.e.  Line 1: 1 is missing, Line 83: survive instead of survival, Line 95: the word marker is doubled, Line 197: miRNAs that are involved.../involved (no capital letter), Line 219/20: hyphen is missing in miR-34a, Line 360: metastatic instead of metastasis).

Reviewer 3 Report

The review article entitled, "Roles of microRNAs in Regulating Cancer Stemness in Head and Neck Cancers" is well constructed and addresses major aspects of miRNA associated with HNSCC stemness where Authors have explicitly reviewed the duplicity of microRNAs as OncomiR or, tumor suppressor properties related to HNSCCs stemness and its key targets and explained its role in anti-cancer therapy. Authors need to address couple of  questions mentioned below :

Since radiotherapy is major therapy for HNSCCs treatment. Did the authors find any study that showed the role/effect of miRNA(s) in radiation response (sensitization/resistance) associated with stemness in HNSCC? PMID: 28515423, PMID: 28325958.

Did the authors observe any differences in the role of non-exosomal vs exosomal (circulating) miRNA and compared their effects on stemness in HNSCC tumors?

Round 2

Reviewer 2 Report

The topic of miRNAs regulation cancer stemness in HNSCC is an emerging and interesting topic. Although extensive re-organization of this review has been performed, it still lacks of readability and structure: (i) Key messages or an outline might help to improve this issue. In many cases the authors just describe, which miRNAs regulate other genes/miRNAs or downstream applications. Thus, the manuscript, the rational of the sentences and the role of the underlying miRNAs is really hard to follow and understand.  (ii) Table 1: it remains unclear which order the listed miRNAs follow. It would be better to re-order this list according to its function. 

Minor issues: L. 54/55 must say laryngeal/oropharyngeal, L. 225/226: it would be better to say: OSCC of the tongue, L. 450: there is an "a" too much (plural or singular), Tbl. 1: (a) HOXA9 must be Italicized, (b) Is miRNA-29b/its family a target of itself? That doesn't make sense.

Round 3

Reviewer 2 Report

The manuscript has significantly improved, however, there are a few errors that are still present and should be corrected prior to publication:

Line 53: Why is tongue named separately? It belongs to oral cavity or oropharynx.

Line 54/55 larynx/oropharyngeal SCCs are completely different from an etiological, therapeutic and prognostic point of view and can’t be combined as LSCC. Also, it should say laryngeal instead of larynx.

Line 66: Should be surgery instead of dissection?

Line 67: surgery can also be combined with both adjuvant chemo- and radiotherapy.

miRNAs in regulating cancer stemness:

Line 198: are surveying miRNAs for diagnosis. Better: “are searching for diagnostic miRNAs”

Line 244/5: This part is plagiarized from the abstract of PMID 30454973 (original sentence in the abstract: “And low expression of miR-196b and high expression of PCDH-17 contributed to an increase in the 5-year-survival rate of LSCC patients.”)

Line 247/8: “migration and EMT” is redundant. The acquisition of migratory potential is one feature of EMT.

Line 250: The sentence doesn’t make sense as it is. MAPK is a kinase not a whole pathway.

Gene names must be Italicized throughout the whole manuscript (Line 255: MYC target 1; Line 256/9: MYCT1…)
